# Exosome-Modified AAV Gene Therapy Attenuates Autoimmune Hepatitis via Enhanced Regulatory T Cell Targeting and Immune Modulation

**DOI:** 10.3390/microorganisms13040823

**Published:** 2025-04-04

**Authors:** Wenwei Shao, Weilin Huang, Yixuan Wang, Helin Sima, Kai Ma, Rongtao Chen, Heqiao Han, Yixuan Yang, Yuchen Bao, Xiaolei Pei, Lei Zhang

**Affiliations:** 1Academy of Medical Engineering and Translational Medicine, Tianjin University, Tianjin 300072, China; hwl1437@163.com (W.H.); hemasilin@163.com (H.S.); 2Medical School, Tianjin University, Tianjin 300072, China; makai3321@163.com (K.M.); c19808632064@126.com (R.C.); hhq_1003191382@tju.edu.cn (H.H.); yangyixuan0127@126.com (Y.Y.); baoyuchen0111@163.com (Y.B.); 3State Key Laboratory of Advanced Medical Materials and Devices, Tianjin University, Tianjin 300072, China; 4State Key Laboratory of Experimental Hematology of China, Institute of Hematology & Blood Diseases Hospital, Chinese Academy of Medical Sciences & Peking Union Medical College, 288 Nanjing Road, Tianjin 300020, China; wangyixuan@ihcams.ac.cn (Y.W.); peixiaolei@ihcams.ac.cn (X.P.)

**Keywords:** autoimmune hepatitis, exosome-associated AAV, Foxp3, regulatory T cells, gene therapy, immune modulation

## Abstract

Autoimmune hepatitis (AIH) is a chronic liver disorder driven by immune dysregulation, marked by reduced regulatory T cells (Tregs) and unchecked inflammation. Current therapies lack specificity and efficacy, necessitating novel approaches. This study explores gene therapy using exosome-associated adeno-associated virus (exo-AAV) to deliver the Foxp3 gene, aiming to restore Treg-mediated immune tolerance in AIH. We engineered exosomes expressing the CD4-targeting antibody on their surface, encapsulating AAV6/Foxp3, to enhance lymphoid cell specificity. In a ConA-induced murine AIH model, engineered exo-AAV administration significantly increased hepatic Treg proportions while reducing Th17 cells and inflammatory cytokines (IFN-γ, TNF-α, IL-6), compared to control groups (unmodified exo-AAV or empty exosomes). Liver histopathology and serum ALT levels also improved in engineered exo-AAV treated mice. Mechanistically, engineered exo-AAV demonstrated superior targeting via CD4 binding, validated by immunofluorescence and nanoparticle tracking. Despite transient reductions in splenic Tregs, localized hepatic immune modulation underscored exo-AAV’s efficacy. These findings highlight engineered exo-AAV as a promising strategy for precision gene therapy in AIH, overcoming limitations of traditional AAV delivery by enhancing lymphocyte-specific transduction and immune balance restoration. This approach presents a novel therapeutic avenue for systemic autoimmune diseases reliant on Treg reinforcement.

## 1. Introduction

Autoimmune hepatitis (AIH) can lead to long-term liver injury and inflammation, hypergammaglobulinemia with elevated IgG levels, increased transaminase values, interfacial hepatitis, and autoantibody overproduction [1,2]. Current estimates suggest an annual diagnosis rate between 100,000 and 200,000 cases in United States. Comparative data from European epidemiological studies reveal lower population metrics, with incidence rates ranging from 0.9 to 2 cases per 100,000 people annually and prevalence rates of 11–25 per 100,000 [3,4]. Although the specific causes of AIH have not been clarified, failure to receive appropriate treatment at an early stage will soon lead to cirrhosis and advanced liver disease [5]. Current studies have revealed that the pathogenesis of AIH involves the molecular simulation interaction between specific genetic traits and disease, and also leads to impaired immune regulation mechanisms, including autoantibodies produced by CD4^+^T cells, Treg cells, cytotoxic CD8^+^T cells and B cells [6]. At present, the treatment of AIH is mostly based on non-specific immunosuppressive drugs, usually a combination of prednisone/prednisone and azathioprine to induce liver inflammation relief [7,8]. Alternative immunosuppressants such as mycophenolate mofetil, cyclosporine, or tacrolimus may be considered in cases of treatment intolerance or resistance. With appropriate treatment, the 10-year survival rate exceeds 90%. However, untreated AIH has a poor prognosis, with a 5-year survival rate below 25% [9]. Although this treatment has a certain effect on most patients, it requires continuous treatment and frequent recurrence. The recurrence rate is as high as 90%, so it is difficult to achieve withdrawal [10]. Between 10 and 20% of patients will progress to cirrhosis and end-stage liver disease, which may require liver transplantation [11,12]. At the same time, lifelong immunosuppression is not without risk, especially in the pediatric population. Studies have shown that after 10 years of treatment, 4% of patients with type 1 AIH eventually develop into hepatocellular carcinoma, with a probability of 2.9% [13].

With the deepening of research, gene therapy has also been used to treat autoimmune diseases. Gene therapy can carry or manipulate genetic materials to treat diseases. It improves the defective genes that lead to the development of diseases and effectively hinders the occurrence or progression of diseases [14]. Adeno-associated virus (AAV), a non-enveloped, single-stranded DNA virus of the genus Dependoparvovirus (family Parvoviridae), exhibits exceptional tropism and safety profiles as a gene delivery vector for therapeutic applications [15]. Globally, AAV-based gene therapies now constitute 8.1% of ongoing clinical trials, reflecting a sustained increase in translational adoption over recent decades. Through surface coupling and encapsulation, capsid modification design and other methods, the limitations of natural AAV itself can be broken through, and the immune response of AAV can also be reduced by means of exosome encapsulation.

AAV demonstrates therapeutic potential in autoimmune disease models. In multiple sclerosis (MS), rAAV vectors encoding myelin oligodendrocyte glycoprotein (MOG) induce antigen-specific Foxp3^+^ T-cell expansion in the liver, bypassing MHC restrictions to alleviate experimental autoimmune encephalomyelitis (EAE), a key MS mode [16,17]. For uveitis, subretinal delivery of AAV2-hIFN-ɑ sustains intraocular human interferon-alpha levels, suppressing experimental autoimmune uveitis (EAU) [18]. In systemic lupus erythematosus (SLE), AAV-mediated expression of TNF-α-inducible protein 8-like 2 (TIPE2) reprograms macrophage polarization toward the M2 phenotype, restoring immune homeostasis and mitigating SLE severity. TIPE2, a regulator of innate and adaptive immunity, counteracts inflammation and tissue damage by balancing M1/M2 macrophage dynamics. These approaches highlight rAAV’s versatility in delivering targeted immune-modulatory genes across diverse autoimmune pathologies, from CNS disorders like MS to systemic conditions such as SLE, offering precision in attenuating disease mechanisms without broad immunosuppression [19].

The majority of AAVs employed in preclinical and clinical studies are confined to natural capsid serotypes, which exhibit significant limitations for widespread clinical application. AAV’s safety profile and tropism engineering capabilities have propelled its use in clinical trials, yet systemic deployment remains hampered by neutralizing antibodies (NAbs) and poor lymphocyte transduction efficiency. NAbs against AAV capsids pre-exist in >50% of the population due to prior viral exposure, and they critically impair therapeutic efficacy by blocking cellular entry, intracellular trafficking, and nuclear translocation of viral particles. To address this, recent advancements in exosome-associated AAVs (exo-AAV) systems have demonstrated enhanced immune evasion [20,21]. The use of exo-AAV for liver-targeted gene delivery can circumvent the host’s humoral immune response against capsid proteins, significantly reduce the required therapeutic vector dose, and enable efficient and safe targeted gene transfer to the liver [22]. By exploiting exosomes’ endogenous lipid bilayer, exo-AAV shields capsid epitopes from NAbs, enabling efficient delivery even in seropositive hosts. In parallel, exosome surface engineering—unlike conventional AAV capsid modifications—allows the incorporation of tissue-specific ligands to enhance targeted delivery. Exosomes (40–150 nm vesicles) are uniquely suited for immune-compatible gene delivery. Their natural composition minimizes off-target effects, while customizable surfaces enable precision targeting. Research has demonstrated that exosome-mediated targeted gene therapy enhances both the safety and efficacy of therapeutic interventions [23]. Exo-AAV demonstrate high biocompatibility, a low clearance rate, and suitability for cell-specific delivery.

Regulatory T lymphocytes (Tregs) constitute essential immunomodulatory elements within adaptive immunity, fundamentally governing immune suppression through the maintenance of self-tolerance and the prevention of pathological hyperresponsiveness. Functional or numerical alterations in Treg populations constitute risk factors for autoimmune pathogenesis. Conventional identification of natural Treg populations primarily relies on co-expression of CD4 and CD25 surface markers, characteristic features of their immunophenotypic signature that demarcates their predominant immunological subset—representing approximately 10% of CD4^+^ T lymphocytes in both murine and human physiological contexts. The current immunological consensus designates Forkhead box P3 (Foxp3), a lineage-defining transcription factor of the forkhead protein family, as the definitive molecular determinant for natural Treg specification and functional competence. When CD4^+^ T cells express Foxp3, they acquire the Treg phenotype, characterized by the expression of surface markers such as CTLA-4 and CD25. Treg cells primarily inhibit the proliferation of effector T cells through two mechanisms: direct cell-to-cell contact leading to suppression, and the production of inhibitory cytokines, such as IL-10 and TGF-β, which exert indirect regulatory effects [24].

Studies have demonstrated a reduction in the number of Treg cells within both the inflammatory infiltrates of AIH tissues and peripheral blood. A series of investigations conducted at King’s College London have indicated that the function of Treg cells in younger AIH patients is compromised, accompanied by a decrease in their numbers. The diminished responsiveness of Treg cells to IL-2 in the peripheral blood of AIH patients results in the impaired production of the anti-inflammatory cytokine IL-10. Grant et al. further observed that Treg cells from AIH patients exhibit heightened sensitivity to IL-2, while their ability to suppress the pro-inflammatory cytokine IL-17 in CD4^+^ T cells is impaired [25]. Th17 cells are a subset of helper T cells that produce IL-17, crucial for defending against extracellular pathogens (e.g., fungi, bacteria). However, overactivation promotes chronic inflammation and autoimmunity (e.g., rheumatoid arthritis) by recruiting neutrophils and disrupting tissue barriers. Defective Treg cell function can lead to the proliferation of cytotoxic T lymphocytes, macrophage and complement activation, as well as the production of pro-inflammatory cytokines by Th1 and Th17 cells. Additionally, alterations in the adhesion of natural killer cells and liver antibody Fc receptor interactions contribute to liver cell damage [26]. Consequently, the pathogenesis of AIH is linked to the loss of immune regulation, and therapeutic strategies should focus on restoring the proliferative capacity of Treg cells and enhancing their numbers.

The presence of neutralizing antibodies in the body and the limited efficiency of infection represents significant challenges in AAV gene therapy. Current AAV serotypes exhibit suboptimal transduction efficiency in lymphocytes. In this study, we engineered an anti-mouse CD4 GK1.5 overexpression plasmid, which was stably expressed on the cell membrane surface of AAV-packaged engineering cells via lentiviral infection. Stable transgenic strains were subsequently screened and purified to obtain exo-AAV capable of specifically binding to CD4 molecules. The Foxp3 gene was introduced using modified exo-AAV, with the aim of enhancing targeting specificity for mouse CD4^+^ T cells. We further investigated whether this approach could mitigate excessive autoimmune responses in vivo, regulate the immune environment, and achieve therapeutic effects in AIH. The therapeutic potential of the exo-AAV carrying the Foxp3 gene was evaluated by comparing liver function markers (ALT and AST), the cell populations in the spleen and liver, and cytokine profiles in treated versus untreated mice. These findings suggest that AIH can be improved at the systemic level, providing a novel strategy for AIH treatment.

## 2. Materials and Methods

### 2.1. Mice

C57BL/6 J mice (aged 8–10 weeks, male) were obtained from Beijing China Vital River Laboratory Animal Technology.

### 2.2. Experimental Design in AIH Mouse Model

The mice were randomly assigned to one of four groups: the ConA control group (which received an equal volume of normal saline), the AAV-Foxp3 group, the EV group, and the GK1.5 group. Animals were systematically assigned to treatment/control groups using a stratified randomization approach. During critical experimental phases, serum ALT quantification, histopathology and flow cytometry analyses were conducted by technicians unaware of group allocations. These groups were injected via the tail vein with solutions containing AAV and exosome-associated AAV (AAV-Foxp3, EV, GK1.5). AAV-Foxp3 (1 × 10^10^ vg, IV) was dosed per established protocols for murine studies. Matching doses of exo-AAV were administered after qPCR quantification of encapsulated genomes. Six days following the injections, a 8 mg/kg solution of ConA was administered via tail vein injection to induce autoimmune hepatitis. Blood samples were collected, and tissue cells were harvested for analysis 12 h post-ConA injection. The number of mice in each group varied due to differences in mortality rates, modeling success rates, and successful cell separation rates.

### 2.3. Histology

For histopathological assessment, liver tissues were fixed, paraffin-embedded, and cut into 5 μm sections prior to H&E staining. Necrotic area quantification was performed on three systematically randomized fields per sample (blinded analysis).

### 2.4. Lentivirus Packaging and Stable Cell Line Construction

293FT cells were seeded in a six-well plate at an appropriate density, achieving 80–90% confluence after 12–16 h to facilitate transfection. Lentivirus was packaged in pairs of wells. For each well, the following reagents were used: 2 μg of target gene plasmid (GK1.5 or EV), 1 μg of the auxiliary plasmid VSVG, 1 μg of PAX2 plasmid, 12 μg of Polyethylenimine (PEI), and 100 μL of Opti-MEM. After 60 h of transfection, the cell culture supernatant containing lentivirus was collected and used to infect AAV293 cells seeded in a 10 cm dish. An additional 10 mL of DMEM was supplemented. After 24 h, the virus-containing medium was replaced with standard cell culture medium. The cells were passaged 48 h later, and 2.5 μg/mL of puromycin was added for selection. The medium was changed daily until the cells showed stable attachment and minimal mortality, resulting in the establishment of AAV293 cell lines stably expressing GK1.5.

### 2.5. Exo-AAV and AAV Preparations

Exo-AAV particles were produced in AAV293 cells. Briefly, a triple transfection was performed using AAV plasmid, rep/cap plasmid (AAV6 serotype), and helper plasmids (pxx680) via PEI. AAV transgene plasmid was constructed with AAV6 inverted terminal repeat (ITR)-flanked mouse Foxp3 (mFoxp3) transgene under the hybrid CMV immediate-early promoter. The medium was replaced with exosome-free DMEM containing 2% FBS the day after transfection. At 72 h post-transfection, the culture medium was collected. To remove cell debris and apoptotic bodies, the supernatant was subjected to sequential centrifugation at 500× *g* for 10 min and 2500× *g* for 10 min. The supernatant containing exo-AAV was then centrifuged at 10,000× *g* for 30 min to remove larger microvesicles. Subsequently, the remaining supernatant was centrifuged at 150,000× *g* for 1 h to pellet the exosome-associated AAV particles. Conventional AAVs were purified from the cell lysate using iodixanol-gradient ultracentrifugation. The purified vectors were stored at −80 °C until use. Prior to titering, the exo-AAV sample was treated with DNase to degrade plasmid DNA from the transfection. This was performed by mixing 2 µL of the sample with 2 µL of DNase I, 2 µL of 10× buffer, and 14 µL of water. The reaction was incubated for 1 h at 37 °C, followed by inactivation of DNase I by heating at 85 °C for 15 min. Finally, the exo-AAV preparations were titrated using quantitative TaqMan PCR, which detects AAV genomes.

### 2.6. Flow Cytometry Analysis

Flow cytometry was employed to assess the expression levels of CD4, CD8, F4/80, Foxp3, IFN-γ, and IL-17A in intrahepatic lymphocytes (IHLs) and splenocytes. IHLs were isolated from liver tissue by centrifugation of single-cell suspensions using Ficoll-Conray (Solarbio, Beijing, China). Intracellular staining was performed using the Fixation/Permeabilization buffer solution (BD Biosciences, San Jose, CA, USA) following the manufacturer’s protocol. Intranuclear staining was similarly conducted using the Fixation/Permeabilization buffer solution (BD Biosciences) as per the manufacturer’s instructions. The following antibodies were used and purchased from Biolegend: FITC anti-mouse CD4 Antibody, APC/Cyanine7 anti-mouse CD8a Antibody, PE/Cyanine7 anti-mouse F4/80 Antibody, PE anti-mouse FOXP3 Antibody, PE Rat IgG2b, κ Isotype Ctrl Antibody, PerCP/Cyanine5.5 anti-mouse IL-17A Antibody, PerCP/Cyanine5.5 Rat IgG1, κ Isotype Ctrl Antibody, APC anti-mouse IFN-γ Antibody, APC Rat IgG1, κ Isotype Ctrl Antibody. The stained cells were analyzed using a CytoFLEX instrument (Beckman Coulter Life Science, Brea, CA, USA).

### 2.7. ELISA

The levels of cytokines, including TNF-α, IFN-γ and IL-6, in murine plasma were detected by ELISA using commercially available kits (CUSABIO) according to the manufacturer’s instruction.

### 2.8. Western Blot Analysis

Cells were lysed using RIPA lysis buffer containing protease and phosphatase inhibitors (Solarbio). Proteins were separated by sodium dodecyl sulfate-polyacrylamide gel electrophoresis (SDS-PAGE) and transferred onto polyvinylidene difluoride (PVDF) membranes. The membranes were incubated overnight at 4 °C with specific primary antibodies: Anti-HA antibody, anti-CD81 antibody, anti-TSG101 antibody, and anti-GAPDH antibody. After washing with Tris-buffered saline containing Tween 20, the membranes were incubated with anti-rabbit secondary antibodies. Protein bands were visualized using an enhanced chemiluminescence (ECL) detection system (Beyotime, Shanghai, China). Quantification of the protein bands was performed using ImageJ (1.8.0) software.

### 2.9. Statistical Analysis

The data in this study were shown as mean ± SEM and analyzed by Graphpad Prism 9 software. The one-way ANOVA with Tukey’s post hoc test was performed. *: *p* < 0.05; **: *p* < 0.01; ***: *p* < 0.001.

## 3. Results

### 3.1. Establishment and Verification of Stable Transfected Cell Line Overexpressing CD4-Binding Protein

GK1.5 is a widely used monoclonal antibody-producing cell line that specifically recognizes mouse CD4^+^ T cells. The antibodies secreted by GK1.5 primarily bind to CD4 molecules through the variable regions of the heavy and light chains. The target gene sequence we designed consists of the CD8α signal peptide, HA tag, GK1.5 heavy chain, linker, GK1.5 light chain, CD8α hinge region, and CD8α transmembrane region, which we refer to as the GK1.5 transgenic construct (Figure 1a). The gene sequences were obtained from NCBI database. The synthesized target gene was cloned into the lentiviral packaging PUE vector via restriction enzyme digestion. Following lentivirus infection and stable strain selection, the transgene was expressed on the cell membrane, enabling purified exosomes to specifically recognize mouse CD4^+^ T cells through the variable regions of the heavy and light chains and bind to CD4 molecules. The corresponding empty vector (EV) sequence was generated by removing the GK1.5 target gene sequence, while retaining all other elements, which was used as control.

After constructing the GK1.5 overexpression lentivirus and the corresponding empty lentivirus, AAV293 cells were infected with these constructs, and monoclonal cell lines were established, resulting in the GK1.5 overexpression stable cell line and the EV cell line. The protein expression of GK1.5 and EV cell lines was subsequently analyzed (Figure 1b). Using the fusion of GK1.5 and an HA-tagged protein in transgenic cells, the expression of the target protein, which is approximately 38 kDa in size, was detected in the GK1.5 stable strain using an HA-tagged antibody, as shown in the figure. Based on these results, the screened stable transgenic strains were confirmed to stably express the target gene.

To confirm the localization of the target protein of GK1.5 on the cell membrane, cell membrane proteins and cytoplasmic proteins were isolated from the GK1.5 stable strain using a cell compartment separation method. These fractions were then compared with whole-cell protein extracts and a positive control. The HA tag, which is carried by the GK1.5 protein, was detected exclusively on the membrane, while the EV control showed no expression (Figure 1c). Additionally, immunofluorescence analysis revealed red fluorescence labeled with GK1.5 in the GK1.5 stable strain. Most of the fluorescence was observed surrounding the nucleus and on the cell membrane. These results demonstrate that the GK1.5 protein is successfully and stably localized on the cell membrane, thus achieving its intended function of specifically binding to mouse CD4 (Figure 1d).

### 3.2. Exo-AAV Identification

To identification the exosomes, the morphology of exosomes was detected by transmission electron microscopy, and adeno-associated virus particles and exosome-associated adeno-associated viruses were observed (Figure 2a). The diameter of AAV particles generally ranges from 20 to 26 nm, with visible particle sizes around 20 nm, meeting the required specifications. Exosomes, on the other hand, typically have diameters ranging from 100 to 200 nm, with a double-layered membrane structure observable via transmission electron microscopy. We utilized GK1.5 or EV stable cell lines, transfected them with AAV-Foxp3 packaging-related plasmids, and purified exosome-AAV particles (exo-GK1.5 or exo-EV) separately from culture supernatant through ultracentrifugation. At the same time, AAV-foxp3 vectors were produced and purified from cell lysate. A significant number of exosomes are visible in the right panel of the image, with their average diameter and shape aligning with expectations. Upon further magnification, exosomes with a diameter of approximately 200 nm can be clearly observed, along with AAV particles displaying a hexagonal structure in the center, confirming the successful production of exosome-associated AAV. Additionally, the size distribution of the exosomes was assessed using a nanoparticle size analyzer, which showed a single peak between 100 and 200 nm, indicating that the exosomes’ average particle size falls within this range and demonstrating their high purity. Next, exosome surface markers (CD81, TSG101) were detected by Western blot (Figure 2b). The results, as shown in the figure, indicate the presence of positive bands for the exosome marker proteins at the expected molecular weights, and the HA tag protein was expressed in GK1.5 exosomes. In contrast, the HA tag protein was absent in the empty exosome preparation, further validating the presence of the targeted protein in the GK1.5 exosomes. Particle size analysis (Figure 2c) confirmed purified exosomes (EV and GK-1.5) exhibit a narrow size range (40–150 nm), excluding apoptotic bodies.

### 3.3. Detection of Exo-AAV Binding Ability In Vitro

AAV-Foxp3, exo-EV, and exo-GK1.5 were co-incubated with mouse spleen cells in vitro at 2000 MOI for 2 h, followed by genomic DNA extraction. The AAV content in the genome was quantified using real-time PCR with ITR primers to assess the binding ability of the different groups to mouse spleen cells. As shown in Figure 3, the binding efficiency of the exosome group was significantly higher than that of the AAV-Foxp3 infection group. Furthermore, the binding ability of the GK1.5 group was markedly greater than that of the EV group. These results suggest that encapsulating AAV in exosomes enhances infection efficiency to some extent, with surface-modified exosomes, particularly those expressing GK1.5, providing a more effective improvement in binding and transduction efficiency.

### 3.4. Examination of Liver Damage in AIH Mice After Exo-AAV Treatment

Following AAV intervention in AIH mice, we first assessed changes in liver injury severity. As shown in Figure 4, some liver specimens exhibited signs of congestion or focal necrosis, indicating liver damage. HE staining results revealed that compared to the control group, the exo-GK1.5 group showed a significant reduction in necrotic liver injury (Figure 4b). Additionally, the liver injury severity in the exosome-treated group was significantly lower than that in the AAV-Foxp3 group. Unexpectedly, however, the EV group exhibited more severe hepatic congestion than the untreated group, which may be associated with the inherent effects of exosomes on inflammatory processes. Serum ALT levels were measured in AIH mice, revealing a slight decrease in ALT levels in the AAV-Foxp3, EV, and GK1.5 groups (Figure 4c). Despite the more severe liver congestion observed in the EV group, ALT levels did not show a significant increase, suggesting that the observed liver damage was not strongly correlated with ALT elevation in this case.

### 3.5. Detection of Immune Cells and Inflammatory Cytokines in AIH Mice Liver After Exo-AAV Treatment

After exo-AAV treatment, the proportion of IHLs of AIH mice were detected. As shown in Figure 5a, the Treg cells in GK1.5 group exhibited an increasing trend compared to the EV group. This result suggests that exo-AAV modified with GK1.5 may have enhanced infection efficiency, leading to an increase in the proportion of Treg cells. Next, we assessed the impact of Treg changes induced by exo-AAV on other AIH-related immune cells. Compared to the control group, the proportion of Th17 cells in the GK1.5 group was significantly reduced (Figure 5b). In contrast, the AAV-Foxp3 and EV treatment groups showed a slight reduction in the proportion of Th17 cells, but this change did not reach statistical significance. Additionally, macrophage-related marker F4/80^+^ expression in liver cells was assessed. A significant reduction in the proportion of F4/80^+^ cells was observed in the GK1.5 group compared to the control group (Figure 5c).

Given that IFN-γ and TNF-α are key mediators of liver injury in the ConA-induced mouse model, the levels of these pro-inflammatory cytokines in serum were measured. The results revealed a significant increase in both IFN-γ and TNF-α levels following ConA injection, when compared to the healthy control group (Figure 5d). In the EV treatment group, IFN-γ levels remained largely unchanged, while TNF-α levels showed only a slight increase. In contrast, both IFN-γ and TNF-α levels were significantly reduced in the GK1.5 group, suggesting that the treatment contributed to alleviating inflammation. Furthermore, serum IL-6 levels, which are known to inhibit the development of Treg cells, were also measured. Although IL-6 levels in the EV group increased slightly compared to the control group, this difference was not statistically significant (Figure 5d). However, in the GK1.5 group, IL-6 levels were significantly reduced compared to the EV group, indicating that the expression of GK1.5 on exosome surfaces improved its targeting and infection efficiency, thus promoting the function of mFoxp3 and reducing the inflammatory response.

### 3.6. Identification of Spleen T Cells in AIH Mice After Exo-AAV Treatment

To investigate the changes in the immune environment following exo-AAV gene therapy, the proportion of lymphocytes in the spleen of AIH mice was assessed. As shown in Figure 6a, no significant difference in the proportion of Treg cells was observed in the AAV-Foxp3 group compared to the control group. However, the exo-AAV treatment group exhibited a significant reduction in Treg cell proportion, with the GK1.5 group showing a more pronounced decrease. Additionally, the proportion of Th17 cells in both the EV and GK1.5 groups decreased significantly (Figure 6b), with the GK1.5 group showing a slightly lower proportion than the EV group, although this difference was not statistically significant. Regarding IFN-γ^+^ cells, a significant reduction in the proportion of IFN-γ^+^ cells was observed in both CD4^+^ and CD8^+^ T cells in the GK1.5 group, with this reduction being slightly more pronounced than in the EV group (Figure 6c,d).

## 4. Discussion

In this study, we developed exosomes modified to specifically bind CD4 molecules, creating engineered exosome-associated AAV vectors carrying the Foxp3 gene for autoimmune hepatitis treatment. Results demonstrated that exosome-associated AAV-Foxp3 significantly alleviated AIH in mice by enhancing Tregs and reducing Th17 cells. Notably, GK1.5-modified exosomes outperformed unmodified exosomes, highlighting their enhanced ability to target and infect T cells. This increased infection efficiency improved Treg proportions and suppressed AIH-related inflammation. In the murine context, the findings underscore the potential of CD4-targeted exosome-AAV systems to deliver Foxp3 effectively, modulating T-cell responses and attenuating autoimmune inflammation. This approach offers a promising strategy for precision gene therapy in AIH and other T cell-mediated autoimmune disorders. These preclinical findings suggest testable hypotheses for human AIH immunomodulation.

AIH is a complex autoimmune disease with multifactorial pathogenesis. Therefore, developing an appropriate animal model that closely mimics human AIH is crucial for advancing research. In recent decades, substantial efforts have been made to establish reliable AIH mouse models [7,8]. Concanavalin A (ConA), a natural lectin derived from the seeds of *Canavalia ensiformis*, has become an essential reagent in immunological research and is widely utilized in the development of animal models. It serves as a common tool to induce immune-mediated liver injury and has been used extensively to replicate the pathogenic characteristics observed in AIH patients [6,20]. The single tail vein injection of ConA is a well-established and widely adopted method for inducing acute autoimmune hepatitis in animal models [27,28,29]. However, the ConA-induced model lacks certain typical clinical features of AIH, such as the presence of autoantibodies, typical interface hepatitis, and progressive liver fibrosis. Therefore, in future studies, it may be valuable to consider further validation through other AIH models. For instance, employing known autoantigens to break immune tolerance could be explored to establish a chronic type 2 AIH model [30,31]. Christen et al. employed adenovirus-mediated expression of the human CYP2D6 gene—an established autoantigen in type 2 AIH [32]. Similarly, Hardtke-Wolenski et al. utilized adenovirus encoding anti-formiminotransferase cyclodeaminase (FTCD) antibodies in NOD mice to simulate autoimmune liver injury via a single-tail vein injection [31]. Future investigations should prioritize these chronic AIH models to validate our findings in a clinically representative framework. Such models recapitulate the progressive nature of human AIH, enabling comprehensive evaluation of AAV-delivered immunomodulatory factors over extended periods. This approach will not only clarify therapeutic mechanisms but also assess treatment durability in mirroring realistic clinical scenarios.

Tregs are pivotal anti-inflammatory mediators that play a critical role in suppressing immune responses and maintaining immune homeostasis. It is widely accepted by researchers that the transcription factor Foxp3 serves as a definitive marker for natural Treg cells. Accumulating evidence suggests that increasing the population of Treg cells can ameliorate AIH [33,34,35], indicating that the delivery of Foxp3 via AAV to enhance Treg proportions may represent a promising therapeutic strategy for AIH. Although AAV6 has demonstrated a modest advantage in infecting immune cells, the transduction efficiency of currently identified AAV serotypes in lymphocytes remains suboptimal. Even at high doses (10^5^–10^6^ MOI), the proportion of infected T cells is less than 1% [34,36]. Exosomes, which are nanoscale vesicles secreted by most cell types, play a crucial role in long-distance intercellular communication and are involved in numerous biological processes. Due to their unique advantages, exosomes have been identified as highly efficient drug delivery vehicles, providing a distinctive approach for transporting various therapeutic agents to target cells. Multiple studies have demonstrated that AAV can be detected within exosomes derived from 293T cell culture medium [37,38]. Furthermore, exosome-associated AAV vectors have been shown to mediate significantly higher transduction efficiency in target organs when delivered to mice compared to conventional AAV vectors [39,40]. A previous study has demonstrated that exosome-associated AAV8 is capable of transducing lymphocytes, including CD4^+^ T cells, CD8^+^ T cells, and B cells. However, it did not conduct a comparative analysis of the transduction efficiency between exo-AAV and conventional, non-enveloped AAV particles [41].

To enhance the transduction efficiency of AAV in T cells, we engineered the surface of exosomes by generating a stable transgenic cell line through lentiviral transduction. This approach enabled the stable expression of the antigen-binding region of the GK1.5 monoclonal antibody, which targets mouse CD4, on the exosomal surface. Subsequently, GK1.5-modified exo-AAV particles were purified from the culture medium of AAV293 cells used for AAV packaging. In this study, we confirmed the localization of GK1.5 on the cell membrane of AAV293 cells and detected the presence of GK1.5 target molecules in the purified exosomal fraction. Furthermore, electron microscopy analysis of the purified exosomes revealed that both unmodified exosomes (exo-EV) and GK1.5-modified exosomes contained AAV viral particles. Owing to the size heterogeneity of exosomes and viral particles, the number of viruses encapsulated within each exosome ranged from several to over a dozen. These findings collectively demonstrate the successful generation of GK1.5-modified exosome-associated AAV vectors carrying the Foxp3 gene. Studies have demonstrated that the transduction efficiency of recombinant AAV vectors in human primary CD4^+^ T lymphocytes in vitro is notably low. Experimental data indicate that the transduction rates for several AAV subtypes are only a few percentage points, often close to zero [42]. Consequently, it is challenging to assess the impact of viral infection on the proportion of Tregs in vitro, particularly given the differences in AAV delivery efficiency between in vivo and in vitro systems. As a result, changes in Treg proportions were not detectable in experiments involving exo-AAV infection in vitro. However, the enhanced transduction capability of exo-AAV was confirmed through quantitative PCR analysis of genomic AAV, demonstrating a significant improvement in infection efficiency. Moreover, genetic engineering modifications to the exo-AAV surface further enhanced its performance, underscoring the potential of this approach for targeted gene delivery.

In the analysis of immune cells from AIH mice treated with exosomes, we observed a notable reduction in Tregs within the spleen, despite the anticipated therapeutic outcome of increased Foxp3 expression and elevated Treg proportions. In contrast, the proportion of Tregs in the liver was significantly elevated in the exosome-treated group, with the GK1.5-modified exosome group demonstrating a markedly higher Treg proportion compared to the EV control group. These findings suggest that GK1.5-modified exosomes can enhance the proportion of Foxp3^+^ T cells within the localized inflammatory environment of the liver. The observed reduction in splenic Tregs may be attributed to the migration of Tregs between peripheral and localized immune microenvironments [43].

Following treatment, a reduction in serum ALT levels was observed in AIH mice across the AAV-Foxp3 group, exo-EV group, and exo-GK1.5 group. However, the decreases in the first two groups did not reach statistical significance. In terms of liver histomorphology, studies have shown that the degree of hepatic congestion is positively correlated with the severity of AIH [44,45]. In our study, the exo-GK1.5 group exhibited significantly less hepatic congestion compared to the EV exosome control group, suggesting that GK1.5 may exert a targeted binding effect, potentially alleviating AIH by increasing the proportion of Tregs in the liver. Nevertheless, the exo-EV group displayed more severe hepatic congestion than the control group. Despite this, based on ALT levels and immune cell profiles, the exo-EV group showed some degree of improvement overall compared to the control group, albeit not significantly. It is hypothesized that certain characteristics of exosomes may influence the disease progression of AIH, potentially exacerbating inflammation and congestion. However, this finding does not contradict the conclusion that modifying exo-AAV with membrane-localized CD4-specific binding proteins can enhance its transduction efficiency in lymphocytes. Further research is necessary to explore and validate the impact of exosomes on AIH.

Our study establishes that GK1.5 exosomes significantly enhance CD4^+^ T-cell-targeted delivery of AAV-Foxp3 in AIH models, driving hepatic Treg expansion while suppressing Th17-driven inflammatory pathways. These findings position the platform as a potential breakthrough for refractory AIH, particularly where standard therapies fail. Critical translational challenges, however, demand resolution: (1) Safety risks including off-target transduction, chronic Treg functional exhaustion, and engineered exosome-triggered immune responses; (2) Biological incompatibilities arising from species-specific GK1.5-CD4 binding affinities (murine vs. human), necessitating humanized model validation; (3) Manufacturing standardization requiring GMP-compliant protocols for scalable production, batch consistency, and stability profiling. To bridge preclinical promise to clinical utility, comparative efficacy studies across AIH phases (acute flare vs. chronic fibrosis) should be considered to optimize therapeutic windows. While this targeted approach demonstrates superior precision over systemic immunomodulation, its clinical translation remains contingent upon resolving these multifaceted biological and technical barriers.

## 5. Conclusions

In this study, exosomes modified with GK1.5 protein demonstrated enhanced binding affinity for CD4^+^ T cells, facilitating improved delivery of exosome-associated AAV vectors encoding Foxp3. This approach was designed to modulate the Treg/Th17 balance and mitigate inflammatory processes in AIH. Experimental results revealed that, compared to untreated AIH controls and EV-modified exosomes, GK1.5-exosomes significantly increased intrahepatic Treg frequencies while reducing Th17 populations. Furthermore, downstream pro-inflammatory cytokines (IFN-γ, TNF-α, IL-6) were partially suppressed. These data suggest that surface-engineered exosomes may serve as a platform to enhance AAV tropism for T cells, transiently ameliorating AIH-associated inflammation via Treg expansion, which make this technology a promising, yet still investigational, therapeutic strategy for AIH.

## Figures and Tables

**Figure 1 microorganisms-13-00823-f001:**
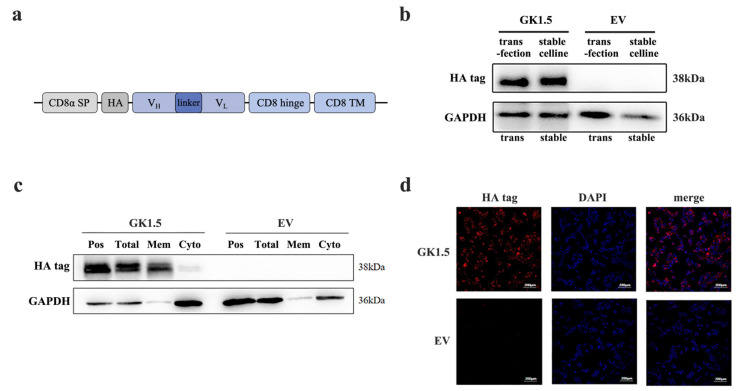
Identification of GK1.5 overexpression stable cell line. (**a**), GK1.5 transgene cassette; (**b**), Identification of the protein level of GK1.5 in the stable overexpression cell line with transient transfection as control; (**c**), Western blot of total protein, membrane protein, cytoplasmic protein and transient plasmid positive control extracted from GK1.5 and EV stably transfected strains by cell component separation; (**d**), GK1.5 was examined by immunofluorescence in GK1.5 and EV stably transfected cells with HA antibody.

**Figure 2 microorganisms-13-00823-f002:**
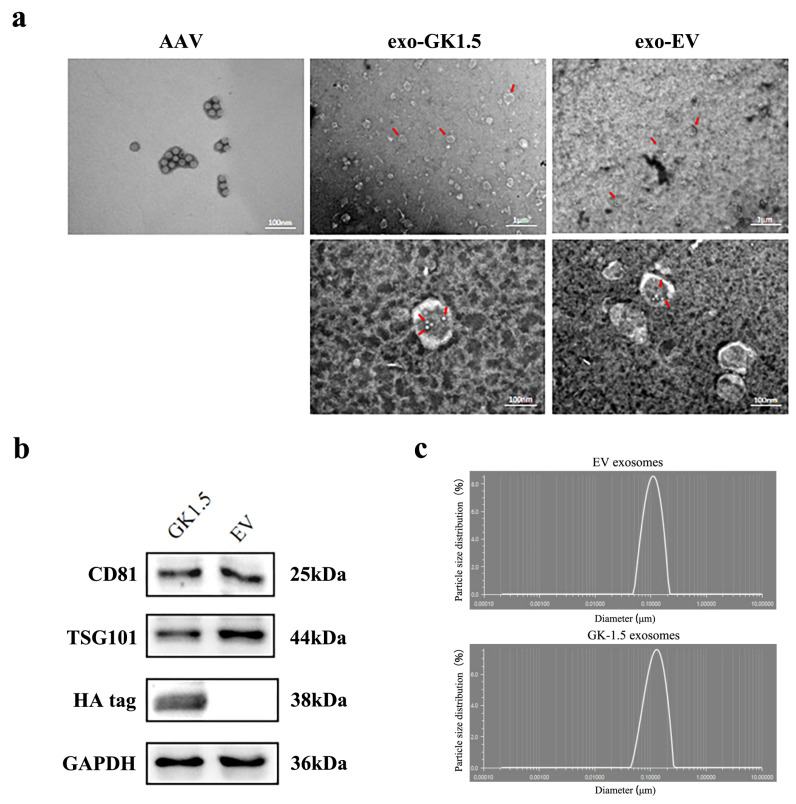
Exosome identification. (**a**), exosome-associated AAV transmission electron microscopy. Red arrow: Exosomes or AAV particles encapsulated therein; (**b**), Exosome particle size test results shown in the nano particle size analyzer. Exosome surface marker proteins (CD81, TSG101) of GK1.5 and EV were detected; (**c**), the particle size of exosomes was detected.

**Figure 3 microorganisms-13-00823-f003:**
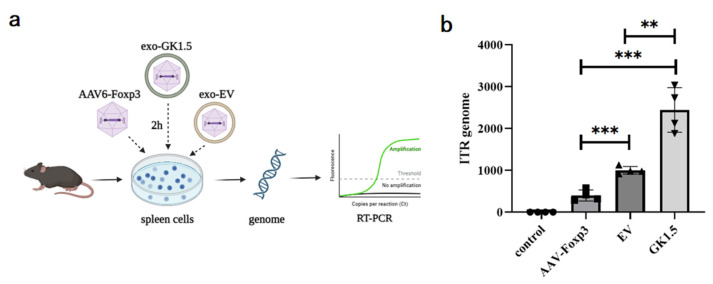
Comparison of exo-AAV binding ability in vitro. (**a**), After AAV-Foxp3, exo-EV and exo-GK1.5 infected mouse splenocytes in vitro for 2 h, the cell genome was extracted for quantitative PCR detection; (**b**), relative AAV genome copy number was analyzed with ITR specific primers. **: *p* < 0.01; ***: *p* < 0.001.

**Figure 4 microorganisms-13-00823-f004:**
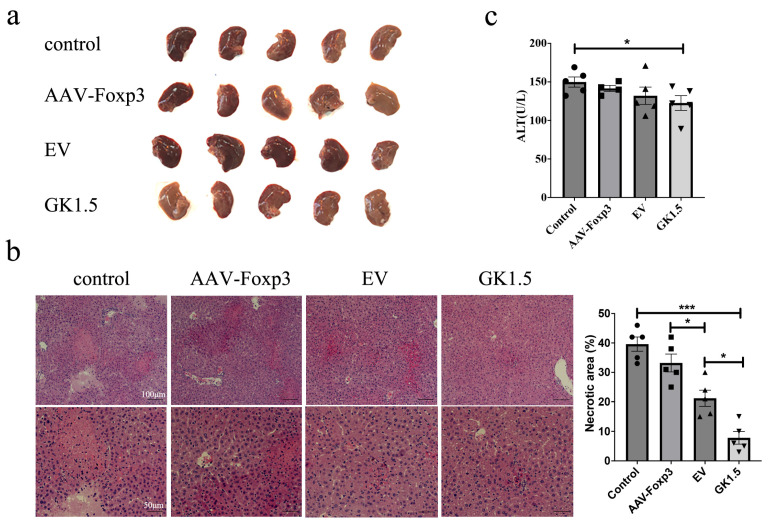
Liver damages after AAV treatment in AIH mouse model. (**a**), the images of mouse liver in different groups of AIH mice. (**b**), H&E staining of liver tissue in different groups of AIH mice, and quantification of necrotic area was presented. (**c**), serum ALT activity was measured. different symbols. *: *p* < 0.05; ***: *p* < 0.001.

**Figure 5 microorganisms-13-00823-f005:**
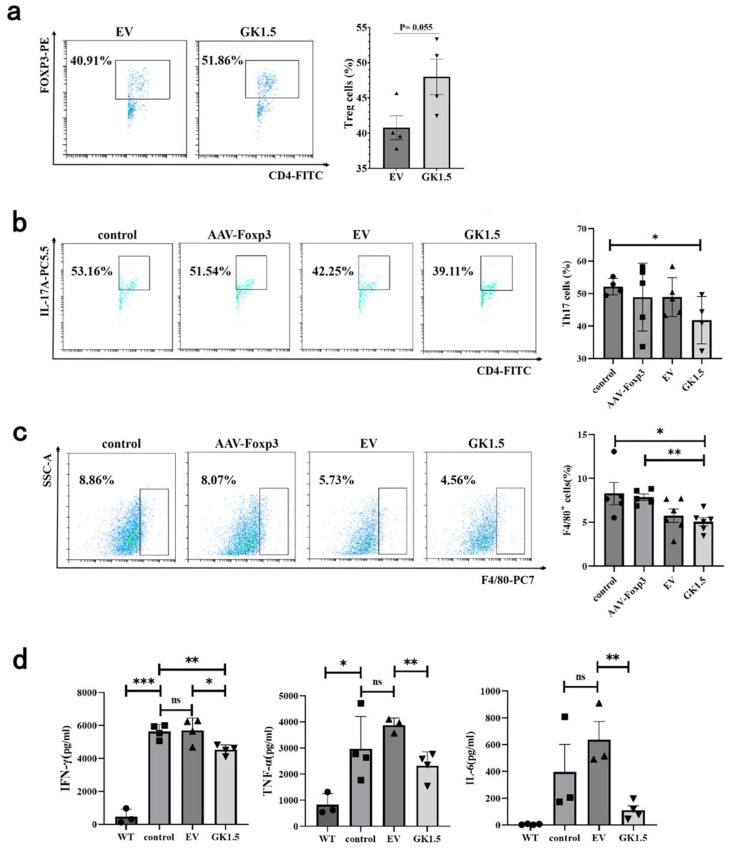
The proportion of liver immune cells and the changes in inflammatory cytokines were detected after exo-AAV treatment. Detection of liver T cells in AIH mice after exo-AAV treatment. AIH mice in different treatment groups were sacrificed 12 h after modeling. Liver cells were taken for flow cytometry and IL-17^+^ and Foxp3^+^ cells were analyzed in the CD4^+^ cell portal to detect the proportion of Treg (**a**) and Th17 (**b**) cells. (**c**), After AAV treatment, the liver macrophages of AIH mice were detected and analyzed, and the changes in F4/80^+^ cell ratio were detected. (**d**), Statistical analysis of serum IFN-γ, TNF-α and IL-6 levels in AIH mice after AAV treatment. *: *p* < 0.05; **: *p* < 0.01; ***: *p* < 0.001; ns: no significance.

**Figure 6 microorganisms-13-00823-f006:**
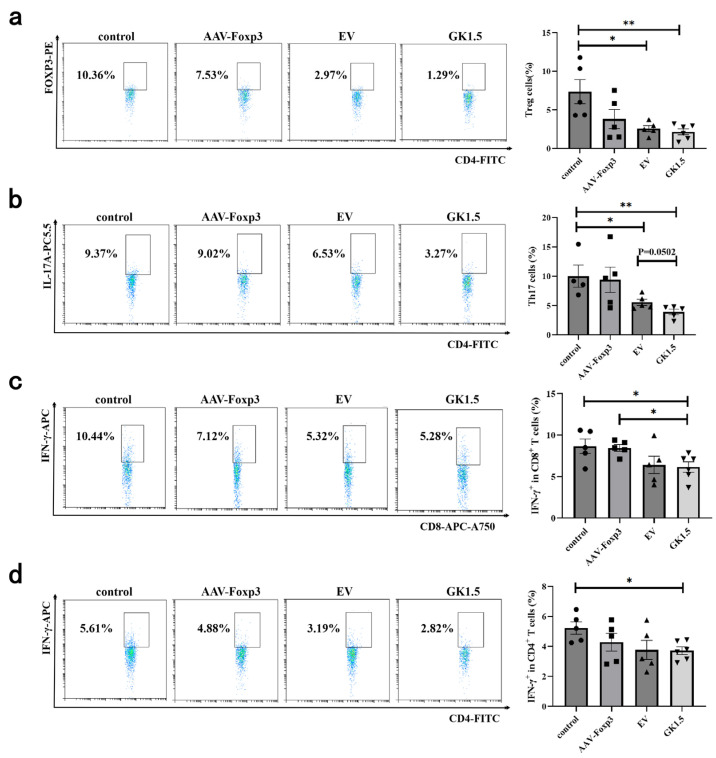
Detection of spleen T cells in AIH mice after exo-AAV treatment. AIH mice in different treatment groups were sacrificed 12 h after modeling. Spleen cells were taken for flow cytometry. Foxp3^+^ and IL-17^+^ cells were analyzed in CD4^+^ cell gates to detect Treg (**a**) and Th17 (**b**) cells. The proportion of IFN-γ^+^ cells was detected in CD8^+^ (**c**) or CD4^+^ (**d**) T cells. *: *p* < 0.05; **: *p* < 0.01.

## Data Availability

The original contributions presented in this study are included in the article. Further inquiries can be directed to the corresponding authors.

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
