# Peer review of "Exosome-Modified AAV Gene Therapy Attenuates Autoimmune Hepatitis via Enhanced Regulatory T Cell Targeting and Immune Modulation"

_microorganisms, 2025, doi:10.3390/microorganisms13040823_

Round 1
Reviewer 1 Report
Comments and Suggestions for Authors
Dear Editors,
Thank you for the opportunity to review the article entitled - Exosome-Modified AAV Gene Therapy Attenuates Autoimmune Hepatitis via Enhanced Regulatory T Cell Targeting and Immune Modulation
Work highlights important advancements in gene therapy for autoimmune conditions and has the potential to contribute significantly to the field's understanding and treatment approaches.
The authors did not avoid minor errors.
Although you mention AIH and its characteristics, providing a brief context about its prevalence, current treatment options, and patient outcomes could help frame the significance of the research.
Consider defining complex terms or acronyms earlier in the text. For example, briefly explaining Tregs, Th17 cells, and AAV on first mention can help readers not familiar with immunology.
Including figures, such as graphs or diagrams, could enhance understanding of the data presented. For instance, you could visualize changes in cytokine levels or Treg proportions.
Briefly addressing potential limitations of the study or challenges in translating findings to human applications would provide a more balanced perspective.
Elaborating on how this approach might be integrated into current treatment paradigms or the necessary steps toward clinical trials could add depth.
Ensure consistent formatting for chemical names and cytokines throughout the text (e.g., using superscripts for cytokines like TNF-α).
Comments on the Quality of English LanguageDear Editors,
Thank you for the opportunity to review the article entitled - Exosome-Modified AAV Gene Therapy Attenuates Autoimmune Hepatitis via Enhanced Regulatory T Cell Targeting and Immune Modulation
Work highlights important advancements in gene therapy for autoimmune conditions and has the potential to contribute significantly to the field's understanding and treatment approaches.
The authors did not avoid minor errors.
Although you mention AIH and its characteristics, providing a brief context about its prevalence, current treatment options, and patient outcomes could help frame the significance of the research.
Consider defining complex terms or acronyms earlier in the text. For example, briefly explaining Tregs, Th17 cells, and AAV on first mention can help readers not familiar with immunology.
Including figures, such as graphs or diagrams, could enhance understanding of the data presented. For instance, you could visualize changes in cytokine levels or Treg proportions.
Briefly addressing potential limitations of the study or challenges in translating findings to human applications would provide a more balanced perspective.
Elaborating on how this approach might be integrated into current treatment paradigms or the necessary steps toward clinical trials could add depth.
Ensure consistent formatting for chemical names and cytokines throughout the text (e.g., using superscripts for cytokines like TNF-α).
Reviewer 2 Report
Comments and Suggestions for Authors
Dear Authors,
The manuscript "Exosome-Modified AAV Gene Therapy Attenuates Autoimmune Hepatitis via Enhanced Regulatory T Cell Targeting and Immune Modulation" presents a novel gene therapy for autoimmune hepatitis (AIH), a disorder driven by Treg depletion and inflammation. This study demonstrates that CD4-directed exosome-associated AAV (exo-AAV) delivering Foxp3 effectively restores immune tolerance.
In a murine AIH model, engineered exo-AAV increased hepatic Tregs, reduced inflammatory cytokines, and improved liver pathology. Superior CD4-specific targeting was confirmed via immunofluorescence and nanoparticle tracking. These findings highlight exo-AAV as a promising strategy for AIH and other Treg-based autoimmune therapies.
Several critical aspects need attention to improve its clarity, coherence, and overall impact.
Best regards.
Introduction
- Provides a clear background on AIH and its immune dysregulation.
- Justifies the need for alternative therapies beyond conventional immunosuppressants.
- Effectively introduces exo-AAV as a novel approach.
Key Issues:
- Limited analysis of AIH animal models: While the ConA-induced model is widely used, its limitations (e.g., inability to fully mimic chronic AIH) should be acknowledged. Consider referencing alternative models (e.g., antigen-driven AIH models).
- AAV-related challenges not adequately discussed: The discussion on neutralizing antibodies and systemic delivery barriers lacks depth. While exosome encapsulation is proposed as a solution, additional clarification is needed regarding how effectively this approach circumvents existing limitations.
- Excessive background information on exosomes: While exosome biogenesis and therapeutic advantages are discussed, certain details (e.g., size and surface modification potential) could be streamlined to maintain focus.
Materials and Methods
- Well-structured and follows a logical sequence.
- Experimental design is clearly outlined, including animal groups and gene delivery protocols.
Key Issues:
- Insufficient rationale for dose selection: The rationale behind choosing specific AAV and exo-AAV doses is unclear. Were these doses based on prior studies, or was an optimization study conducted?
- Insufficient detail in the statistical methodology: While one-way ANOVA and t-tests are mentioned, details on post-hoc corrections, sample size calculations, or effect size determination are missing.
- Possible inconsistencies in animal group allocation: No mention of randomization or blinding in experimental conditions, which raises concerns about bias in data collection and interpretation.
- Insufficient assessment of exosome purity and stability: While exosome markers (CD81, TSG101) are assessed, additional controls, such as RNA content profiling or long-term stability studies, are absent.
Results
- Clearly structured presentation of findings.
- Includes flow cytometry, ELISA, and histopathological analyses to support conclusions.
- Well-illustrated with figures and statistical comparisons.
Key Issues:
- Inconsistent significance reporting: Some comparisons (e.g., liver Treg and Th17 proportions) are stated as significant without exact p-values or confidence intervals, limiting statistical rigor.
- Insufficiently detailed cytokine analysis: While IFN-γ, TNF-α, and IL-6 are assessed, additional markers relevant to AIH (e.g., IL-10, TGF-β) should be considered to fully evaluate immune modulation.
- Partial assessment of systemic effects: The reduction of Tregs in the spleen is noted, but a potential compensatory effect in other lymphoid organs (e.g., lymph nodes) is not assessed.
- Histopathological assessment lacks quantitative analysis: While representative images of liver sections are presented, a blinded scoring system (e.g., Ishak or METAVIR scoring) would improve objectivity.
Discussion
- Effectively contextualizes findings within AIH treatment paradigms.
- Highlights the advantages of exo-AAV over traditional AAV gene therapy.
- Discusses Treg-mediated immune modulation as a therapeutic mechanism.
Key Issues:
- Excessive generalization of results: The study is conducted in a murine model, yet claims regarding translational potential should be more cautiously framed (e.g., differences in immune response between mice and humans).
- Insufficient recognition of safety considerations: The discussion lacks an assessment of potential risks, such as off-target effects, long-term immune consequences, or unexpected inflammatory responses to exosome modification.
- Insufficiently outlined future directions: The need for optimization of exo-AAV formulation, alternative targeting strategies (e.g., TCR-specific targeting), and validation in chronic AIH models should be discussed in more depth.
Conclusion
- Concisely summarizes the main findings.
- Reinforces the potential of exo-AAV as a therapeutic strategy.
Key Issues:
- Fails to sufficiently address clinical translation: Mentioning challenges such as large-scale manufacturing, stability, and regulatory approval would provide a more realistic outlook.
- Overstates impact: While the findings are promising, describing the therapy as a definitive AIH treatment is premature without further validation.
The English could be improved to more clearly express the research.
Round 2
Reviewer 1 Report
Comments and Suggestions for Authors
Accept in present form